# Thermal Behaviour of Rebars and Steel Deck Components of Composite Slabs under Natural Fire

**Marcílio M. A. Filho** [1] , **Paulo A. G. Piloto** [2] **and Carlos Balsa** [3,*]

1    ISISE, Department of Civil Engineering, Universidade do Minho, Campus Azurém,
     4800-058 Guimarães, Portugal
2    LAETA/INEGI, Instituto Politécnico de Bragança, Campus Santa Apolónia, 5300-253 Bragança, Portugal
3    Research Centre in Digitalization and Intelligent Robotics (CeDRI), Instituto Politécnico de Bragança,
     Campus Santa Apolónia, 5300-253 Bragança, Portugal
*    Correspondence: balsa@ipb.pt

**Abstract:** Most of the studies involving composite slabs under fire follow the standard fire scenario described by the ISO 834 curve, disregarding the cooling-phase. However, recent studies show that this phase is equally important, as it can lead to the collapse of the structure. Therefore, the present research carried out a parametric study, using numerical models, validated through experimental tests, to evaluate the thermal behaviour of the composite slabs components under natural fire. The results showed that the maximum temperatures in the reinforcement bars occur during the cooling-phase, reaching temperatures up to 300% higher than at the heating-phase, on the steel deck occur at the end of heating, and that the concrete thickness above the steel deck influences the temperature of these components. Also, during the cooling-phase, a "heat bubble" effect is observed on the ribs of the composite slabs, where the reinforcement bars are normally placed. These results highlight the importance of considering different natural fire scenarios, in the structural performance and safety of composite slabs, since during the cooling-phase there is still heat transfer between the elements, which can lead to slab failure. New parameters are proposed to find the temperature of each component for different fire ratings.

**Keywords:** natural fire; fire resistance; composite slab; steel deck; numerical model

## 1. Introduction

The composite slabs with steel deck present themselves as a very efficient and lightweight structural solution compared with conventional reinforced concrete slabs. Considering the advantages presented, this slab model has become widely disseminated in building projects based on steel structures and tall buildings. According to Yu, X. et al. [1], the slabs consist of a thin cold rolled steel sheet, commonly with thicknesses between 0.6 and 1.2 mm, the concrete being thrown over the steel plate profile. The concrete is usually reinforced with a light anti-crack mesh and can also include reinforcement bars, usually positioned on the ribs.

The existence of an unprotected steel deck in the composite slabs also presents a disadvantage concerning the fire resistance. Considering that the steel deck can be directly exposed to fire, its structural stability can be affected, because the increase in temperature reduces strength and stiffness [2]. The problem occurs because the steel components are directly responsible for the load bearing of the composite slabs, which play a fundamental role in the stability of the structure during the fire, avoiding also the spread of fire to other compartments.

According to Gillie et al. [3], one of the main objectives in fire safety engineering, is to prevent the structure from catastrophic collapse during a fire event, ensuring that the structure withstands long enough for occupants to evacuate safely. Jiang et al. [4] mention that normally the fire resistance classification of structural elements is based on fire tests in

which the element is submitted during a period of time to a standard fire curve, usually ISO 834 [5]. The furnace temperature is pre-set using an equation, which allows finding the fire resistance according to load bearing (R), insulation (I), and integrity (E).

Jiang et al. [6] highlight that the simplified formulations proposed by Eurocode 4 [7] overestimate the fire resistance, besides the moisture content has a significant effect on the temperature distribution in composite slabs. In order to analyse the ultimate strength behaviour of composite slabs under fire conditions, Wu et al. [8] test nine composite slabs with different variables, finding that the thickness of the slabs significantly affects the final load bearing capacity. Cirpici et al. [9] use finite element-based numerical models to simulate composite slabs subjected to standard fire, so that fire was considered on the upper face of the slab, on the concrete floor, with the heat flow departing from the concrete to the steel deck. In this study, it was found that passive fire protection materials offer higher resistance even after 60 min of fast fire exposure.

However, recent studies point out the importance of considering the effects of the cooling phase on the strength of structural elements. Full-scale tests carried out between 1995 and 2003 by the Building Research Establishment (BRE) [10,11], found that failures in the frame connections usually occur during the cooling phase. In addition, another important full-scale test was conducted in 2008 in the Czech Republic [12], showing that the composite slab of an administrative building lost compressive strength during the cooling phase, leading to the collapse of the structure. Gernay and Franssen [13] warn about the risks of disregarding the cooling phase in current approaches to structural fire engineering since the possibility of late failure during or after cooling cannot be ruled out. More recently Ramesh et al. [14] and Choe et al. [15] concluded that failure of the connections between composite beams and columns may trigger a partial or total collapse of the slab and the continuity between slabs can lead to crack propagation in the negative moment regions, which can lead to the fire propagation between floors.

Therefore, it becomes essential to study and understand the thermal behaviour and the load bearing capacity of this element when submitted to different fire events. This study presents the assessment of the temperature component on the steel deck and on the rebars of the composite slabs, using different types of geometry and different types of fire scenarios, characterized by a heating and cooling phase. Numerical simulations are presented with MATLAB and ANSYS programs, seeking to increase the reliability of the results by observing the convergence between both numerical models. The numerical validation was performed with results obtained from experimental tests performed by other authors.

## 2. Materials and Methods

This investigation aims to propose new parameters that encompass natural fire in the determination of the temperature of the components from the composite slabs. The formulations applied to standard fire proposed by EN 1994-1-2 [7] were used as a basis of the new proposal. The thermal simulations used to obtain the data were based on three-dimensional finite element models and validated through the experimental results presented by Guo and Bailey [16,17]. The parametric study included a total of 128 thermal simulations performed in ANSYS and MATLAB software.

### 2.1. Fire Resistance

The load bearing criterion (R) reflects the capacity of the structural element to withstand the load during the fire without collapsing or suffering excessive displacement or rate of displacement. The Annex D of EN 1994-1-2 [7] provides a simplified calculation method for this criterion. This method is based on formulas to determine the temperature of each component in the steel deck: upper flange, lower flange, web, and on the rebars. These are the main elements to resist the tensile stresses resulting from the analysis for positive moment (sagging moment). For the calculation of the temperature in the respective regions

of the steel deck ($\theta_a$), this standard presents the following equation, only valid for standard ISO 834 [5] fire.

$$\theta_a = b_0 + b_1 \frac{1}{l_3} + b_2 \frac{A}{L_r} + b_3 \phi + b_4 \phi^2 \tag{1}$$

The $b_i$ parameters are defined according to Table 1 and depend on the type of concrete used in the structural element and on the region of the steel deck to be analysed. This temperature depends on the steel deck geometry and on the upper flange view factor.

**Table 1.** Coefficients for the determination of the temperature in steel deck for normal weight concrete [7].

| Fire Resistance (min) | Part of the Steel Deck | $b_0$ (°C) | $b_1$ (°C.mm) | $b_2$ (°C.mm) | $b_3$ (°C) | $b_4$ (°C) |
|---|---|---|---|---|---|---|
| | Lower flange | 951 | −1197 | −2.32 | 86.4 | −150.7 |
| 60 | Web | 661 | −833 | −2.96 | 537.7 | −351.9 |
| | Upper flange | 340 | −3269 | −2.62 | 1148.4 | −679.8 |
| | Lower flange | 1018 | −839 | −1.55 | 65.1 | −108.1 |
| 90 | Web | 816 | −959 | −2.21 | 464.9 | −340.2 |
| | Upper flange | 618 | −2786 | −1.79 | 767.9 | −472 |
| | Lower flange | 1063 | −679 | −1.13 | 46.7 | −82.8 |
| 120 | Web | 925 | −949 | −1.82 | 344.2 | −267.4 |
| | Upper flange | 770 | −2460 | −1.67 | 592.6 | −379 |

The view factor ($\phi$) is the coefficient responsible for determining how much radiation the exposed surface receives. Having in mind the numerical simulations, special considerations should be pointed out. For the lower flange, the view factor is considered $\phi_{low,f} = 1$, because it is parallel and directly exposed to fire. However, the other components are not directly exposed to fire, and their average view factors should be determined by the Equations (2) and (3), according to Hottel crossed-strings method [18].

$$\phi_{upp,f} = \frac{\sqrt{h_2^2 + \left(l_3 + \frac{l_1 - l_2}{2}\right)^2} - \sqrt{h_2^2 + \left(\frac{l_1 - l_2}{2}\right)^2}}{l_3} \tag{2}$$

$$\phi_{web} = \frac{\sqrt{h_2^2 + \left(\frac{l_1 - l_2}{2}\right)^2} + (l_3 + l_1 - l_2) - \sqrt{h_2^2 + \left(l_3 + \frac{l_1 - l_2}{2}\right)^2}}{2\sqrt{h_2^2 + \left(\frac{l_1 - l_2}{2}\right)^2}} \tag{3}$$

For the temperature in rebars ($\theta_s$), the EN1994 part 1.2 [7] presents Equation (4), only valid for standard ISO 834 fire. The $c_i$ parameters are shown in Table 2 and depend on the type of concrete used in the structure, also depends on the geometry of the steel deck, on the distance $u_3$ with respect to the lower flange and on the position of the rebar relative to the rib ($u_1$ and $u_2$).

$$\theta_s = c_0 + c_1 \frac{u_3}{h_2} + c_2 z + c_3 \frac{A}{L_r} + c_4 \alpha + c_5 \frac{1}{l_3} \tag{4}$$

Figure 1 shows the geometric parameters used for the determination of the view factor $\phi$.

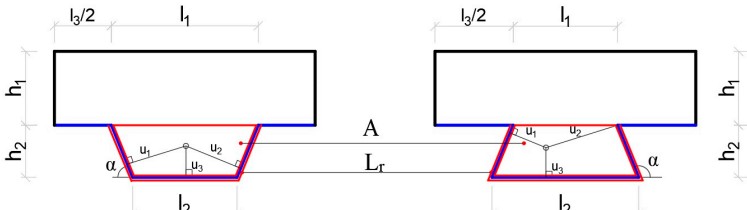

**Figure 1.** Definition of the geometry factors for the composite slab: $l_1$, $l_2$, $l_3$, $h_1$ and $h_2$ (mm) are structure geometry parameters; $A$ (mm$^3$/m) is the concrete volume of the rib per meter of rib length; $Lr$ (mm$^2$/m) is the exposed area of the rib per meter of rib length; $u_1$ and $u_2$ (mm) are defined by the shortest distance of the centre of the rebar to any point of the webs of the steel deck; $u_3$ (mm) is the distance to the lower flange and $\alpha$ (°) is the angle of the web.

**Table 2.** Coefficients for the determination of the temperature in rebars [7].

| Fire Resistance (min) | $c_0$ (°C) | $c_1$ (°C) | $c_2$ (°C.mm$^{-0.5}$) | $c_3$ (°C.mm) | $c_4$ (°C/°) | $c_5$ (°C.mm) |
|---|---|---|---|---|---|---|
| 60 | 1191 | −250 | −240 | −5.01 | 1.04 | −925 |
| 90 | 1342 | −256 | −235 | −5.30 | 1.39 | −1267 |
| 120 | 1387 | −238 | −227 | −4.79 | 1.68 | −1326 |

*2.2. Numerical Model and Validation*

The numerical models were developed by two finite element programs. MATLAB uses linear tetrahedron finite elements with four nodes to generate the mesh of each region (volume). ANSYS uses a combination of different linear finite elements. Concrete is modelled by the hexahedron solid 3D element (SOLID70) with 8 nodes. The steel deck is modelled by finite shell elements (SHELL131) with four nodes, suitable for modelling thin-walled structures. The rebars are modelled using unidimensional elements (LINK33) with two nodes.

The boundary conditions include the recommendations from the EN 1991-1-2 [19]. In the lower part of the slab, boundary conditions correspond to heat transfer by convection and radiation and in the upper part to convection and radiation. The other four surfaces of the slab (front, back, left and right) are considered adiabatic.

It should also be noted that the thermal properties of the materials follow the recommendation of the Eurocodes EN 1992-1-2 [20], EN 1993-1-2 [21] and EN 1994-1-2 [7]. Additional thermal properties have been used for air [18] to model the air gap created by the debonding effect between the steel deck and the concrete.

In order to verify the reliability and accuracy of the numerical results, some real tests of composite slabs under natural fire are simulated. Guo and Bailey [16,17], carried out real tests of composite slabs using a natural fire scenario. These authors present seven tests with composite slabs made with steel deck submitted to different load levels and fire events. Three tests were used for the validation models. The composite slabs were made by normal weight concrete using the trapezoidal steel deck model (CF60/1.2). The steel mesh (A193) was made using 7 mm diameter bars spaced each 200 mm. The height of the concrete was $h_1 = 85$ mm over the steel deck and the thickness of the plate was 1.2 mm, as represented in Figure 2.

The fire duration used in the simulations is 300 min, a little less than the time reported by the other authors. In the numerical models, an air gap layer is considered between the steel deck and the concrete with a constant thickness of 0.5 mm, as suggested by Piloto et al. [22]. During the fire exposure of the composite slab, the steel deck heats up rapidly expanding, thus generating a separation from the concrete. This debonding effect is going to be identified with the name of "air-gap". Sharma et al. [23] also performed numerical analyses using ABAQUS program to study the "air-gap" effect. The authors found that the thermal insulation effect caused by the "air-gap" layer is fundamental for an

accurate prediction of the concrete temperature, and can significantly change the predicted fire resistance rating of the composite slab.

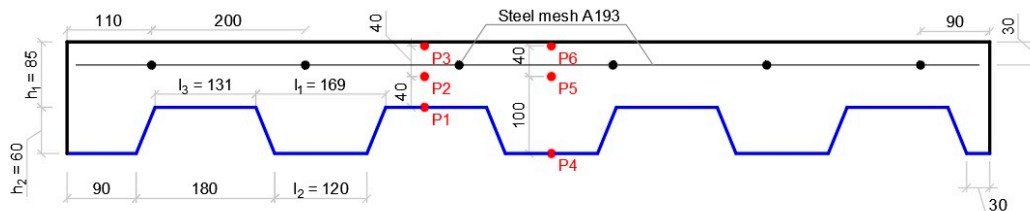

**Figure 2.** Geometric arrangements of the tested composite slabs (Adapted with permission from Guo and Bailey [16]). Red dots (P1-P6) represent the position of the thermocouples, Black dots represent the steel mesh.

Three different fire scenarios with heating and cooling stages are simulated, being the temperature inside the furnace represented in Figure 3. Fire Type 1 and Fire Type 3 have the same heating rate and stage duration (50 min), but the cooling rate of the Fire Type 1 is higher. The heating stage of the Fire Type 2 is 100 min, and the maximum temperature is lower than in the other two fire scenarios. The cooling rate of the Fire Type 2 is similar to the Fire Type 1. All the fire scenarios have two main heating rates, but Fire Type 2 has the second heating stage with a smaller heating rate.

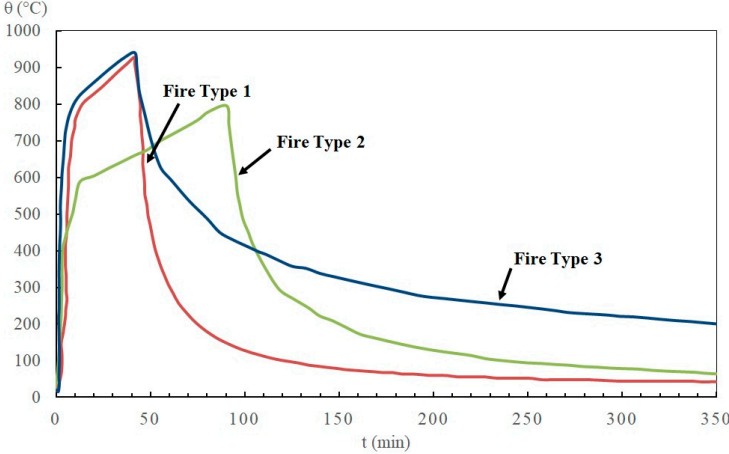

**Figure 3.** Fire scenarios validation model (Adapted with permission from Guo and Bailey [16]).

The characteristics of the simulated slabs used for the validation model are presented in the Table 3. The compressive strength and the moisture content are different for all the slabs.

**Table 3.** Characteristics of the tested composite slabs used for validation.

| Slab N° | Concrete Strength (MPa) | Fire Scenario | Moisture (%) |
| --- | --- | --- | --- |
| Slab 1 | 36.4 | Fire Type 1 | 1.50 |
| Slab 2 | 37.8 | Fire Type 2 | 2.00 |
| Slab 3 | 21.1 | Fire Type 3 | 1.86 |

*2.3. Parametric Study*

A total of 128 thermal simulations were performed for this parametric study, trying to capture the effect of the fire scenario, the effect of different geometries and concrete thickness. A total of 64 simulations are developed using ANSYS and 64 simulations are developed using MATLAB.

The reinforcement of the slabs is the same for all the composite slabs, using 3 steel rebars S500 with a diameter Ø10 mm, always located at a height ($h_2$) from the exposed

surface. The reinforcement mesh uses a steel grid S500 with bars with a diameter Ø6 mm, spaced every 150 mm and positioned 20 mm below the unexposed surface of the slab. The concrete grade used for the models is based on the normal weight type C25/30 with 2% moisture. The composite slab presents the length of 3 m and a width of 1 m.

The parametric analysis is defined by four different commercial profiles of steel deck (Multideck 50, Bondeck, Polideck 59S and Confraplus 60) as shown in the Figure 4, using four different concrete thickness ($h_1$) above the steel deck (40, 60, 80 and 100 mm) and four different natural fire curves (Parametric *i*). Three fire curves present a heating rate similar to standard fire ISO 834 curve (parameter $\Gamma$ closer to 1), and one other curve with higher heating rate, as shown in Figure 5 and Table 4.

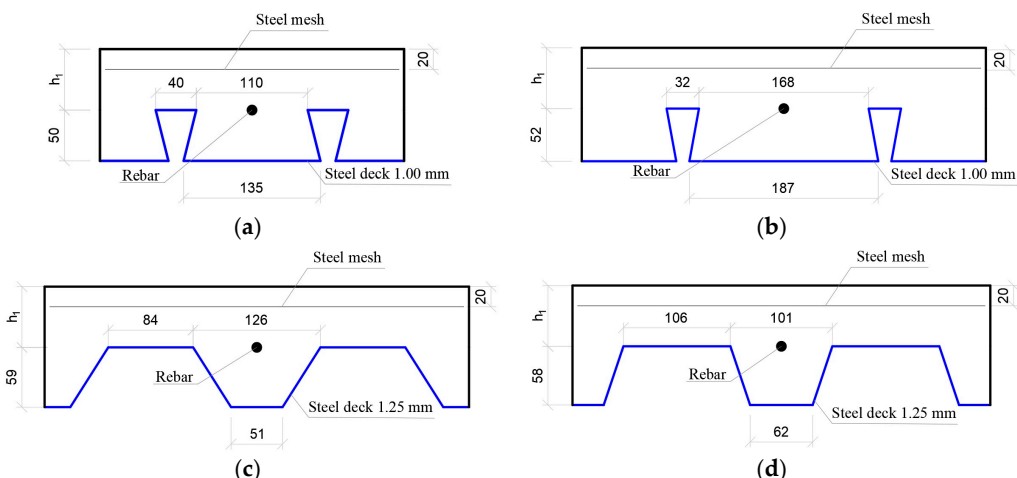

**Figure 4.** Commercial steel deck profiles used in the parametric study for different values of $h_1$, with measures in millimetres: (**a**) Multideck 50; (**b**) Bondeck; (**c**) Polideck 59S; (**d**) Confraplus 60.

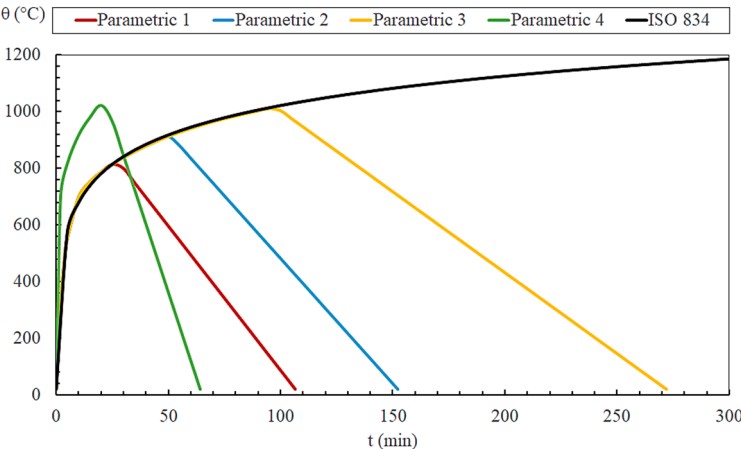

**Figure 5.** Parametric fire curves.

**Table 4.** Data referring to parametric fire curves.

| Fire Scenario | $O$ (m$^{1/2}$) | $q_d$ (MJ/m$^2$) | Maximum Temperature (°C) | Time for the Maximum Temperature (min) | Fire Duration (min) |
|---|---|---|---|---|---|
| Parametric 1 | 0.0465 | 106.32 | 825.79 | 27.43 | 106 |
| Parametric 2 | 0.0465 | 197.47 | 915.38 | 50.93 | 152 |
| Parametric 3 | 0.0465 | 379.75 | 1015.05 | 98 | 272 |
| Parametric 4 | 0.1074 | 197.47 | 1036.60 | 22 | 64 |

The parametric fire curves were elaborated through formulations proposed by EN1991-1-2 [19] for the determination of the parametric temperature-time curves, representing the heating and cooling phases. The model detailed in Annex A of the standard, contains physical limitations on the compartment to be considered, these are: the floor area must not exceed 500 m$^2$; there must be no openings in the compartment ceiling and the maximum height is 4 m.

The heating phase is calculated according to Equation (5), and the cooling phase is represented in the parametric curves by a decreasing linear relationship between temperature and time, and should be calculated according to Equations (6)–(8). Where $\Theta_g$ is the gas temperature (°C) in the fire compartment.

$$\Theta_g = 20 + 1325\left(1 - 0.324e^{-0.2t^*} - 0.204e^{-1.7t^*} - 0.472e^{-19t^*}\right) \tag{5}$$

$$\Theta_g = \Theta_{max} - 625(t^* - t^*_{max}\, x) \text{ for } t^*_{max} \leq 0.5 \tag{6}$$

$$\Theta_g = \Theta_{max} - 250(3 - t^*_{max})(t^* - t^*_{max}\, x) \text{ for } 0.5 < t^*_{max} \leq 2 \tag{7}$$

$$\Theta_g = \Theta_{max} - 250(t^* - t^*_{max}\, x) \text{ for } t^*_{max} > 2 \tag{8}$$

The four curves considered the same compartment dimensions, with the same fencing typology, so that only the opening factor $O$ (m$^{1/2}$), and fire load density $q_d$ (MJ/m$^2$) values were modified, as presented in Table 4.

It can be seen that the increase in the opening factor produces an increase in the maximum temperature, which occurs more rapidly, as well as a decrease in the duration of the fire. Nevertheless, the increase in the fire load density, implies an increase in the maximum temperature, which this time happens later, also increasing the fire duration time.

## 3. Results

The results obtained from the validation of the numerical models, the parametric study, as well as the new coefficients proposed for determining the temperature in the composite slab components will be presented below.

### 3.1. Numerical Validation

Figure 6 shows the comparison between the temperatures determined for Slab 1, obtained from the experimental tests and from the numerical results, using the programs ANSYS and MATLAB. The points where the temperature curves are collected are represented in Figure 2. The results of the Slab 1 agree very well. Similar results were obtained for the other simulated slabs. The difference between the numerical results is related to the finite element type and the mesh size. For both programs, the mesh size was selected based on a convergence test of the solution.

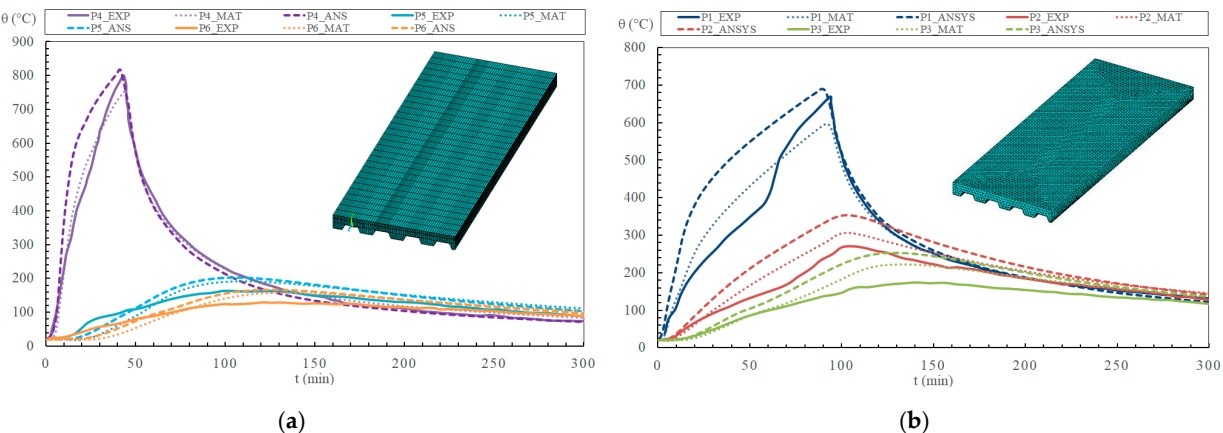

**(a)**           **(b)**

**Figure 6.** Numerical model validation using MATLAB and ANSYS: (**a**) Points P1, P2 and P3; (**b**) Points P4, P5 and P6.

The air gap size is an average constant value, valid through all the thermal simulation. The debonding effect is an event due to the existence of different thermal expansion coefficients between steel and concrete. The real size of the air gap in experiments is expected to increase, depending on the fire event. The results obtained through the two numerical programs overestimate the temperatures in the concrete region.

The Table 5 presents the root mean square error (RMS) observed between the numerical models and the experimental results, during all the simulation periods. This value reflects a good approximation, considering that, at any time after the first 10 min of any standard fire test, the temperature recorded by any thermocouple in the furnace shall not differ from the corresponding temperature of the standard temperature/time curve by more than 100 °C [24].

**Table 5.** Root mean square error (°C) between experimental and numerical results.

| Temperature Location | RMS ANSYS (°C) | Number of Data Analysed | RMS MATLAB (°C) | Number of Data Analysed |
|---|---|---|---|---|
| P1 | 28.95 | 86 | 19.41 | 87 |
| P2 | 20.04 | 60 | 23.50 | 60 |
| P3 | 27.84 | 60 | 30.40 | 60 |
| P4 | 19.61 | 90 | 9.98 | 90 |
| P5 | 21.33 | 58 | 25.31 | 59 |
| P6 | 22.57 | 61 | 26.56 | 69 |

*3.2. Parametric Study*

The Figure 7 presents the temperature field when using the fire curve Parametric 2 and 3. These results are represented for a time equal to 100 min, which belongs to the cooling stage for the Parametric fire curve 2 and to the heating stage for Parametric fire curve 3. The results are determined for a concrete thickness of $h_1 = 60$ mm.

In addition, Figure 8 represents the temperature history of specific points, for all different types of composite slabs, represented for the same concrete thickness $h_1 = 60$ mm, when submitted to the same Parametric fire curve 2 and 3. Similar results are expected for both fire scenarios up to the time 50.93 min, where both heating curves are coincident.

The numerical results obtained with ANSYS were evaluated for every component of the composite slabs. The average temperature has been recorded for the lower flange, web, and upper flange. New coefficients $b_i$ and $c_i$ are presented for the determination of the average temperatures in the steel deck components and in the rebars, respectively, considering composite slabs under these fire events.

The proposed coefficients were obtained in the MS Excel software, which minimized the sum of the absolute differences between the numerical temperatures, obtained in the steel components in the parametric study, and the temperatures obtained by the model corresponding to Equation (1) or (4). This sum was minimized using the Generalized Reduced Gradient (GRG) nonlinear solver [25], included in the SOLVER tool of MS Excel. The tool changed the values of $b_i$ and $c_i$ iteratively aiming to achieve the smallest possible absolute error between the results obtained by the equations, using the new coefficients and the numerical results.

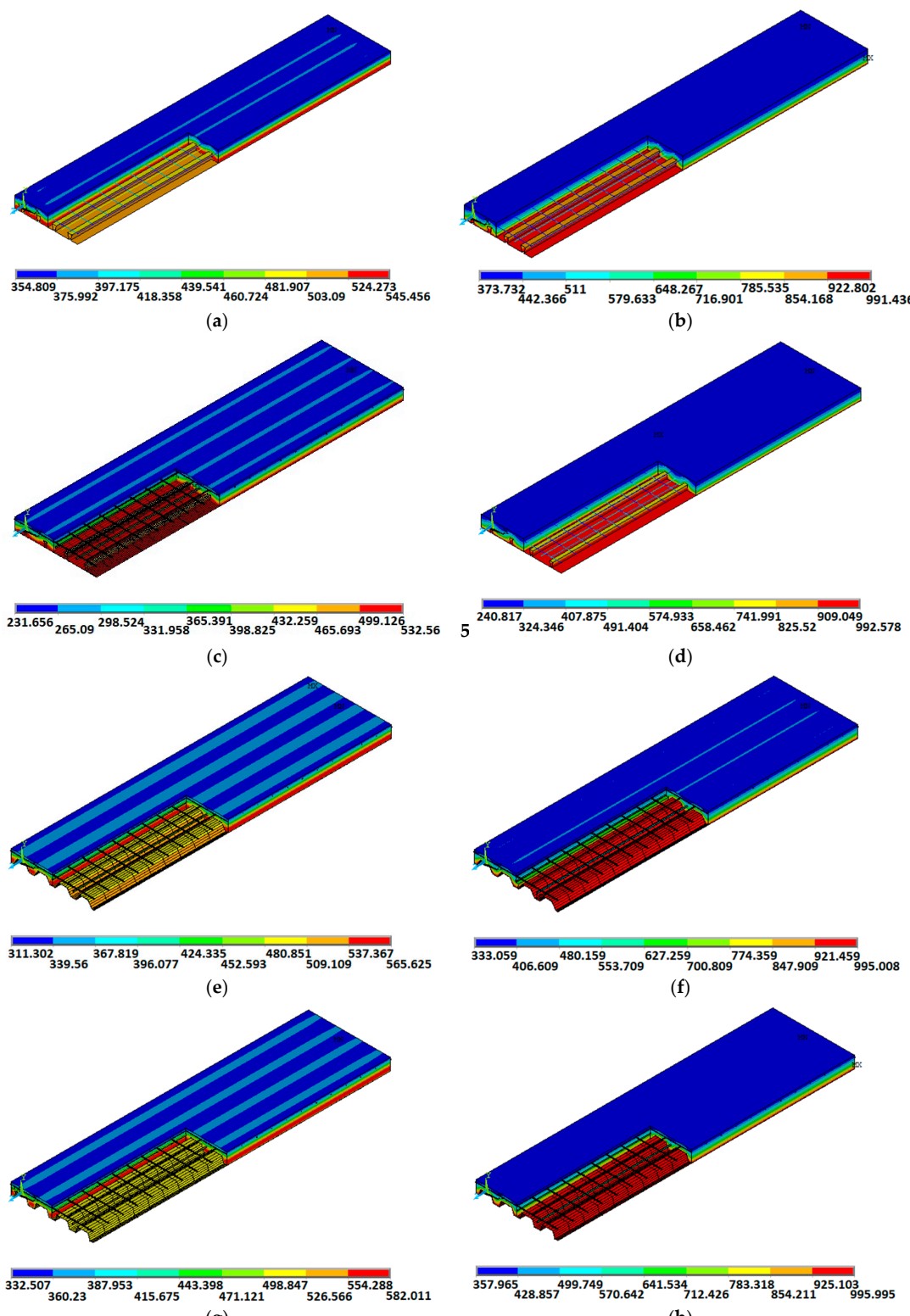

**Figure 7.** Temperature field over the composite slabs under parametric fire curves for the same $h_1$ = 60 mm and for time equal to 100 min: (**a**) Multideck 50 under parametric fire 2; (**b**) Multideck 50 under parametric fire 3; (**c**) Bondeck under parametric fire 2; (**d**) Bondeck under parametric fire 3; (**e**) Polideck 59S under parametric fire 2; (**f**) Polideck 59S under parametric fire 3; (**g**) Confraplus 60 under parametric fire 2; (**h**) Confraplus 60 under parametric fire 3.

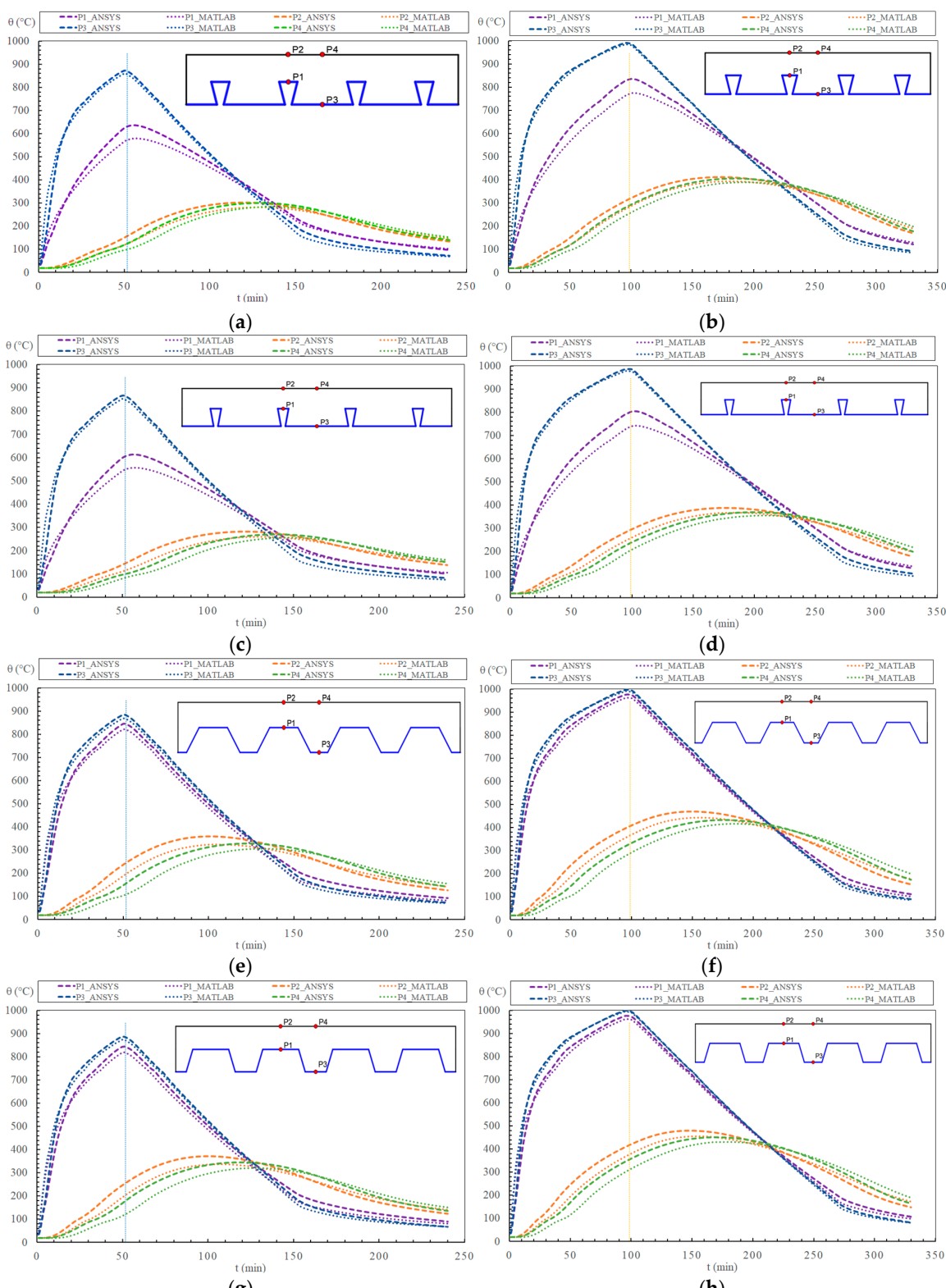

**Figure 8.** Temperature history in key points of the composite slabs under parametric fire curves for the same $h_1 = 60$ mm: (**a**) Multideck 50 under parametric fire 2; (**b**) Multideck 50 under parametric fire 3; (**c**) Bondeck under parametric fire 2; (**d**) Bondeck under parametric fire 3; (**e**) Polideck 59S under parametric fire 2; (**f**) Polideck 59S under parametric fire 3; (**g**) Confraplus 60 under parametric fire 2; (**h**) Confraplus 60 under parametric fire 3.

The Table 6 presents some $b_i$ coefficients proposed to determine the temperature in the steel deck components, and keep using Equation (1). It should be noted that, during the parametric study, the average temperatures in the steel deck and in the rebars are decreasing, whenever there is an increase in the thickness $h_1$. Therefore, the coefficients are proposed in accordance with the thickness $h_1$. The full list of coefficients can be obtained from the research work of Filho, M. [26].

**Table 6.** New coefficients proposed to determine the temperature in steel deck under fire curve Parametric 4 and for the fire rating of 60 min.

| Steel Deck Region | $h_1$ (mm) | $b_0$ (°C) | $b_1$ (°C.mm) | $b_2$ (°C.mm) | $b_3$ (°C) | $b_4$ (°C) |
|---|---|---|---|---|---|---|
| Lower flange | 40 | 417.49 | −919.60 | −0.44 | 82.58 | −162.31 |
| | 60 | 277.50 | −894.68 | −0.29 | 117.52 | −66.23 |
| | 80 | 287.26 | −862.63 | −0.30 | 104.91 | −64.79 |
| | 100 | 366.69 | −859.17 | −0.30 | 59.66 | −99.15 |
| Web | 40 | 319.79 | −461.57 | 0.07 | 109.57 | −145.35 |
| | 60 | 303.15 | −311.20 | 0.08 | 123.14 | −158.80 |
| | 80 | 296.47 | −184.24 | 0.08 | 125.76 | −161.77 |
| | 100 | 295.63 | −179.00 | 0.09 | 123.51 | −159.15 |
| Upper flange | 40 | 333.36 | −832.67 | −0.03 | 37.84 | −56.32 |
| | 60 | 294.20 | −402.71 | −0.02 | 66.74 | −78.82 |
| | 80 | 275.97 | −18.78 | −0.02 | 75.34 | −87.19 |
| | 100 | 275.90 | −17.50 | −0.02 | 57.11 | −68.48 |

The Table 7 presents the convergence of the temperatures observed in the steel deck components of the numerical model and the temperatures obtained in the simplified method using the new coefficients $b_i$ proposed in Table 6.

**Table 7.** Comparison of the temperatures of the numerical model and the simplified method on the steel deck under fire curve Parametric 4 and for the fire rating of 60 min, using the coefficients proposed.

| Profile | Lower Flange | | | Web | | | Upper Flange | | |
|---|---|---|---|---|---|---|---|---|---|
| | ANSYS $\theta$ (°C) | Simplified $\theta$ (°C) | Relative Error (%) | ANSYS $\theta$ (°C) | Simplified $\theta$ (°C) | Relative Error (%) | ANSYS $\theta$ (°C) | Simplified $\theta$ (°C) | Relative Error (%) |
| Multideck 50 | 303.53 | 303.53 | 0.00% | 318.99 | 318.99 | 0.00% | 316.03 | 316.03 | 0.00% |
| Bondeck | 295.24 | 295.24 | 0.00% | 316.13 | 316.13 | 0.00% | 310.23 | 310.23 | 0.00% |
| Polideck 59S | 314.85 | 314.85 | 0.00% | 324.38 | 324.38 | 0.00% | 318.21 | 318.21 | 0.00% |
| Confraplus 60 | 318.91 | 317.88 | 0.32% | 333.37 | 333.36 | 0.00% | 322.59 | 322.59 | 0.00% |

In order to demonstrate the reliability of the new coefficients, Figure 9 presents the temperatures of the steel deck components for the Confraplus 60 profile under the Parametric 4 fire curve, obtained with the ANSYS thermal simulations and with the new proposal.

The Table 8 presents the coefficients that should be used with Equation (4) to determine the temperature in the rebars. They also depend on the value of the concrete cover $h_1$.

The Figure 10 represents the difference between the effect of the standard fire and the parametric fires, confirming the efficiency of the ANSYS numerical model and the new $c_i$ coefficients shown in Table 8.

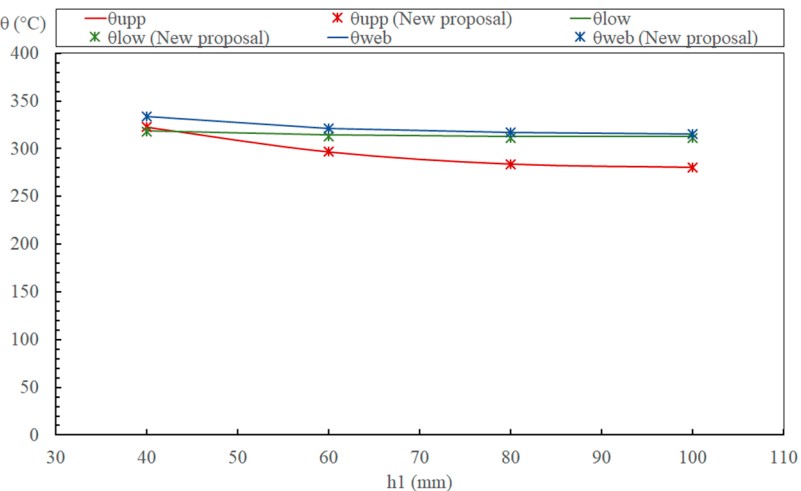

**Figure 9.** Temperature of the steel deck components for Confraplus 60 under Parametric fire curve 4, for the fire rating period of 60 min. (ANSYS results and new proposal).

**Table 8.** New coefficients proposed to determine the temperature in the rebars under the fire curve Parametric 3.

| Fire Resistance (min) | $h_1$ (mm) | $c_0$ (°C) | $c_1$ (°C) | $c_2$ (°C.mm$^{-0.5}$) | $c_3$ (°C.mm) | $c_4$ (°C/°) | $c_5$ (°C.mm) |
|---|---|---|---|---|---|---|---|
| | 40 | 975.25 | −255.43 | −5.63 | −4.69 | 1.23 | −8131.27 |
| 90 | 60 | 922.77 | −257.43 | −5.58 | −4.2 | 1.28 | −8109.41 |
| | 80 | 882.37 | −258.24 | −5.52 | −3.63 | 1.34 | −7955.37 |
| | 100 | 875.9 | −258.54 | −5.53 | −3.7 | 1.34 | −7785.69 |
| | 40 | 1113.95 | −238.19 | −35.51 | −4.34 | 1.07 | −7286.5 |
| 120 | 60 | 1059.87 | −239.67 | −33.88 | −3.82 | 1.17 | −7675.97 |
| | 80 | 1014.32 | −240.6 | −31.77 | −3.36 | 1.25 | −7735.01 |
| | 100 | 994.27 | −241.03 | −31.14 | −3.25 | 1.28 | −7597.36 |

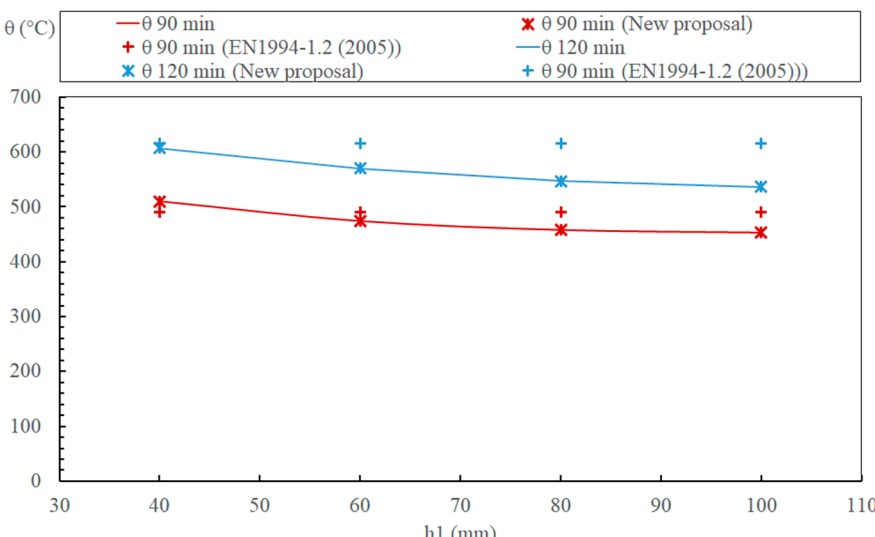

**Figure 10.** Temperature of the rebar component for Multideck 50 under Parametric fire curve 3 and standard fire, for the fire rating period of 90 and 120 min. (ANSYS results and new proposal).

In order to better demonstrate the "heat bubble" effect previously mentioned, Figure 11 presents the temperature profile obtained in ANSYS during the cooling phase of the composite slab with Multideck 50 profile.

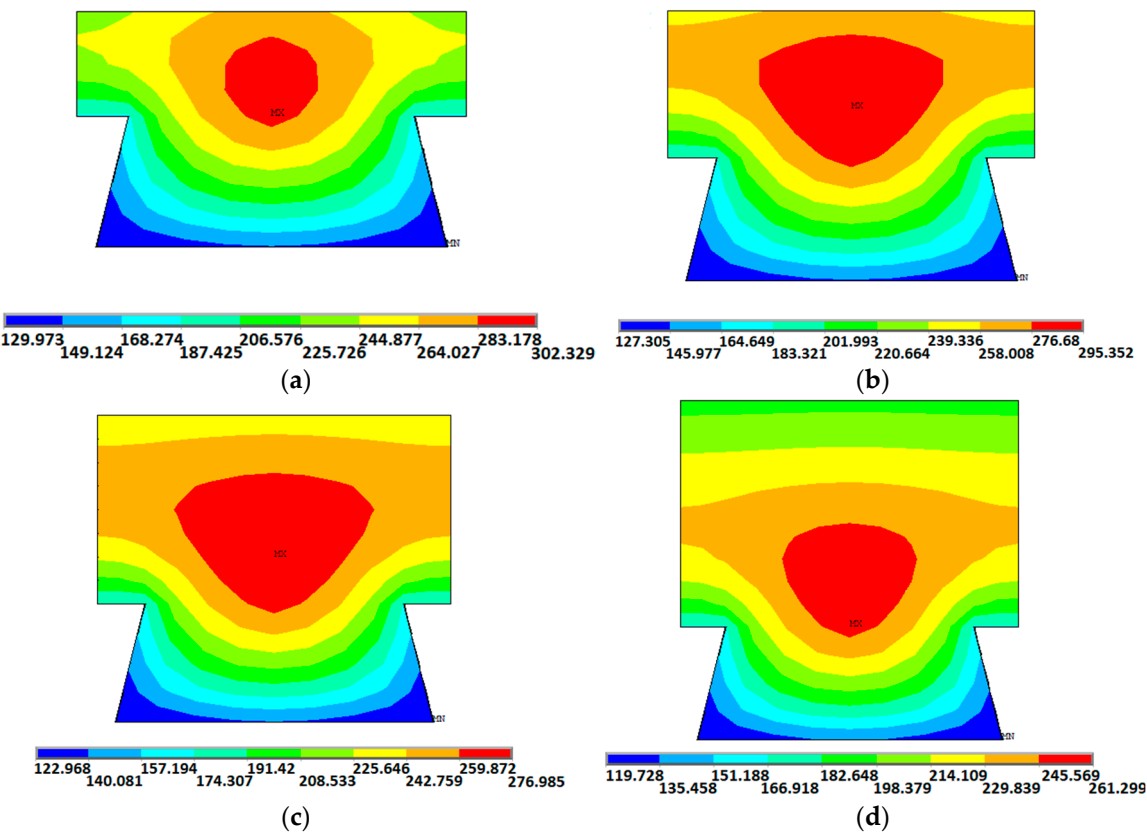

**Figure 11.** Temperature profile during the cooling phase of the composite slab Multideck 50 for different values of $h_1$: (**a**) $h_1 = 40$ mm; (**b**) $h_1 = 60$ mm; (**c**) $h_1 = 80$ mm; (**d**) $h_1 = 100$ mm.

## 4. Discussion

According to the data presented in Section 3, this section will discuss the reliability of the numerical model by comparing the experimental and the numerical results, as well as the reliability and application of the new proposal.

### 4.1. Numerical Validation

According to the curves shown in Figure 6 and the RMS presents in Table 5, the results obtained with MATLAB are closer to the experimental results, especially at points P1 and P4, which are located on the steel deck. On the other hand, the prediction for the maximum temperature on these points is better approximated by ANSYS. The numerical results for Point P3 and P6, which are located only 5 mm from the unexposed side, show a small difference in the initial stage of the fire. In the initial phase of the fire, the simulations underestimate the experimental temperatures, except for points P1 and P4, while in the cooling phase, the simulations end up overestimating the measured temperature in the concrete.

The results obtained with ANSYS overestimate the temperatures in all regions of the composite slab, including the steel deck. Such effect can be explained by the model used to simulate the effect of the air gap. The steel deck and the air gap are represented as solid finite elements in MATLAB, while in ANSYS they are represented by two layers of shell finite elements, using coincident nodes.

In addition to the different finite elements used for modelling composite slabs in the ANSYS and MATLAB software, it's worth noting that the MATLAB models did not contain the steel mesh and the rebars. This effect can also justify the difference in results between both models, since the steel components have high thermal exposure, their presence has an important role on the temperature field of the slabs.

In general, the numerical models present a good approximation for the temperature field on the composite slab as a whole, when using different heating and cooling rates.

More experimental tests will be required in order to better calibrate the model and reduce the difference in temperature values observed in numerical models. However, it is worth mentioning the scarcity of results of experimental tests with composite slabs under natural fire, since the tests are very expensive and most of the existing studies are based on the standard fire curve ISO 834.

*4.2. Parametric Study*

The excellent results obtained in Table 7 indicate that the Equations (1) and (4) proposed by the standard EN 1994-1-2 [7], can be used to obtain temperatures in the steel components of composite slabs subjected to natural fire, provided that appropriate $b_i$ and $c_i$ coefficients are used. It's also noteworthy the fact that the equations only allow the estimation of the temperature in the steel components for the specific fire ratings of 30, 60 and 120 min, making it impossible to describe the temperature evolution throughout the fire scenario. In this way, different $bi$ and $c_i$ coefficients can be assigned for the given resistance times, independent of the fire phase.

It is important to emphasize that the fire duration of the Parametric curve 4 is 64 min, and after 64 min the gas temperature is kept at 20 °C. So, at the end of the fire, the temperatures of the components are still above 150 °C, see Figure 9.

It was also found that the temperatures in the web and upper flange decrease with the concrete thickness $h_1$, reaching a relative difference of 13.04% between the temperatures in the upper flange for $h_1$ = 40 and 100 mm. However, the temperature at the lower flange remains approximately the same for all concrete thicknesses.

According to the results presented in Figure 10, it should be noted that for a time equal to 60 and 90 min, the Parametric fire curve 3 presents similar results to the standard fire curve ISO 834, during the heating stage. Therefore, the behaviour observed in the rebars, for these fire ratings in the numerical model is similar to the results obtained with the simplified method of Eurocode.

In addition, the temperatures are higher in the cooling phase. This effect in the cooling phase can justified by the "heat bubble" generated in the region of the ribs, as shown in Figure 11. It is worth mentioning the fact that the heat zone effect in the region of the rebars was observed for all models during the cooling phase, as well as the fact that the highest temperatures in this component were observed during this phase of the fire event.

**5. Conclusions**

Three-dimensional models have been developed and validated, based on experimental results. These models include the effect of debonding between the steel deck and concrete. This effect has been modeled with solid and shell finite elements to increase the accuracy of the numerical results. There is a small difference between the numerical results and the experimental results, having a root mean square error below 31 °C, and this difference can be attributed to the types and shapes of the finite elements used, as well as the presence of the reinforcement mesh and reinforcement bars in the ANSYS models, which are not included in the MATLAB models.

A parametric analysis has been developed to determine the average temperature of the steel deck components and the temperature of rebars. The temperature of these components affects the reduction factors, which will be responsible for the reduction of the bending resistance under fire (sagging moment). This parametric analysis considered 4 types of composite slabs, 4 types of parametric fire events and 4 types for the concrete thickness ($h_1$).

New coefficients are proposed to find the fire resistance of composite slabs under parametric fire events, the coefficients were able to predict the temperatures obtained in the simulation with a relative error below 0.50%. The new coefficients take into consideration the thickness of the concrete $h_1$, since for slabs with $h_1$ of 40 and 100 mm, the relative difference between the temperatures at the rebars and steel deck can reach up to 20 and 23% respectively, such that the largest differences were observed at the upper flange. This

$h_1$ parameter influences the temperature of the components, which is not considered in the current version of EN 1994-1-2.

This research demonstrates the possibility of extending the simplified model of EN 1994-1-2 to determine the temperature in the components of composite slabs to natural fire. However, for the method to become robust and applicable to the design, it is necessary to obtain a larger database with more natural fire curves, different steel deck profiles, mechanical simulations and experimental tests to assess different potential failure modes, as mentioned by Ramesh et al. [14] and Choe et al. 2019 [15].

In addition, the generation of the "heat bubble" was observed during the cooling stage in the ribs of all the tested models, with temperatures in this region higher than those observed on the exposed and unexposed surfaces. The high temperature of this bubble is affecting the mechanical resistance of rebars during the cooling phase, which can compromise its load bearing behaviour, increasing the fire risk, considering that the effects after the end of the heating phase are usually neglected by the standards and designers.

This investigation highlights the possibility of the collapse during the cooling phase of the fire event and the lack of experimental tests with parametric fire or natural fires.

**Author Contributions:** Conceptualization, P.A.G.P. and C.B.; methodology, P.A.G.P. and C.B.; software, M.M.A.F., P.A.G.P. and C.B.; validation, M.M.A.F.; formal analysis, M.M.A.F.; investigation, M.M.A.F.; resources, P.A.G.P. and C.B.; data curation, M.M.A.F.; writing—original draft preparation, M.M.A.F., P.A.G.P. and C.B.; writing—review and editing, M.M.A.F., P.A.G.P. and C.B.; visualization, M.M.A.F., P.A.G.P. and C.B.; supervision, P.A.G.P. and C.B.; project administration, P.A.G.P. and C.B.; funding acquisition, P.A.G.P. and C.B. All authors have read and agreed to the published version of the manuscript.

**Funding:** This research received no external funding.

**Institutional Review Board Statement:** Not applicable.

**Informed Consent Statement:** Not applicable.

**Data Availability Statement:** Data available upon request to the correspondant author.

**Acknowledgments:** Authors are thankful to their universities for supporting the computational facilities.

**Conflicts of Interest:** The authors declare no conflict of interest.

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
