# Peer review of "Thermal Behaviour of Rebars and Steel Deck Components of Composite Slabs under Natural Fire"

_jcs, doi:10.3390/jcs6080232_

Round 1
Reviewer 1 Report
1. The caption of Fig. 1 need to be expanded to explain the different symbols used in the figure. In summary, the figure and the caption needs to be self explanatory.
2. Figure 3: Fire cannot be counted in the same way as composite slabs. So, instead of using terminologies: Fire 1, Fire 2 and Fire 3, it can be represented as Fire Type 1, Fire Type 2 or Fire Type 3. or, the word degree or mode may be used in between Fire and number, to describe its characteristics.
3. Please increase the font size of the labels used in scale bars of simulation figures.
4. The information must be presented more quantitatively in abstract and conclusions .
Reviewer 2 Report
The topic of this paper concerning the thermal performance of a composite floor slab exposed to fire is valuable to the scientific community. The authors correctly point out that the cooling phase is often neglected and plays and important role in evaluating potential failure mechanisms.
Overall, the conclusions and technical value of this paper is limited. One must consider mechanical loading and end constraint conditions to properly evaluate failure mechanisms and potential for collapse. Simple thermal analysis can provide temperate fields from a given heating profile, but that is a relatively easy first step to predicting mechanical behavior.
The new aspect of this work is the simple model based on parameter fitting, however the validation of this model is insufficient. In Table 7 and Figure 9 the results of temperature predictions from the simplified model are compared to the ANSYS finite element model. The table shows an error of 0.00% for most points, however this is misleading since the parameters are determined from the ANSYS results. It would be more informative to show the statistical R value of the regression analysis.
I’m not convinced by the authors claim that the “heat bubble” plays a significant role is reducing the structural integrity of the floor during the cooling phase. An experimental study conducted by NIST (see reference below) determined excessing stress developed in end connections during the cooling phase and ultimately led to failure. The NIST study provides rich data for model validation and is freely available for download.
Choe, L. et al NIST Technical Note 2055 Compartment Fire Experiments on Long-Span Composite-Beams with Simple Shear Connections. 2019
Round 2
Reviewer 1 Report
I recommend for the acceptance of the manuscript.
Reviewer 2 Report
Authors addressed majority of my original review comments